# A Hierarchical Stabilization Control Method for a Three-Axis Gimbal Based on Sea–Sky-Line Detection

**DOI:** 10.3390/s22072587

**Published:** 2022-03-28

**Authors:** Zhanhua Xin, Shihan Kong, Yuquan Wu, Gan Zhan, Junzhi Yu

**Affiliations:** 1Department of Advanced Manufacturing and Robotics, College of Engineering, Peking University, Beijing 100871, China; huashen18713361816@gmail.com (Z.X.); kongshihan@pku.edu.cn (S.K.); 2Institute of Software, Chinese Academy of Sciences, Beijing 100190, China; yuquan@iscas.ac.cn; 3School of Mechatronical Engineering, Beijing Institute of Technology, Beijing 100081, China; 3120160145@bit.edu.cn

**Keywords:** video stabilization, surface vehicles, three-DOF gimbal, sea–sky-line detection, attitude angle solution

## Abstract

Obtaining a stable video sequence for cameras on surface vehicles is always a challenging problem due to the severe disturbances in heavy sea environments. Aiming at this problem, this paper proposes a novel hierarchical stabilization method based on real-time sea–sky-line detection. More specifically, a hierarchical image stabilization control method that combines mechanical image stabilization with electronic image stabilization is adopted. With respect to the mechanical image stabilization method, a gimbal with three degrees of freedom (DOFs) and with a robust controller is utilized for the primary motion compensation. In addition, the electronic image stabilization method based on sea–sky-line detection in video sequences accomplishes motion estimation and compensation. The Canny algorithm and Hough transform are utilized to detect the sea–sky line. Noticeably, an image-clipping strategy based on prior information is implemented to ensure real-time performance, which can effectively improve the processing speed and reduce the equipment performance requirements. The experimental results indicate that the proposed method for mechanical and electronic stabilization can reduce the vibration by 74.2% and 42.1%, respectively.

## 1. Introduction

As global marine activities continue to heat up, military and civilian marine activities dramatically increase, resulting in the great growth of the use of marine videos. In the civilian field, conventional offshore operations, exploration, and search and rescue often require the assistance of maritime videos. In the military field, visual information, as the main method of passive perception, also has significant value. However, in the process of video capturing, there will be severe disturbances due to wave fluctuations and mechanical vibrations of the camera platform, which will reduce the quality of marine videos and considerably affect the subsequent processing and application [1]. In addition, for storage and transmission, stable marine videos can substantially improve the video compression ratio, which is conducive to saving storage space, speeding up the transmission process, and obtaining higher image quality in the same code stream [2].

Video stabilization is an effective way to solve the problems from disturbances. However, there are still many difficulties. On the one hand, in the case of a heavy sea environment, various noise signals with different frequencies are superimposed because of the violent vibrations. The visual environment is complex and ever-changing, and it often lacks stable feature points, bringing great difficulties to the stabilization process. On the other hand, the motion of surface video includes not only the subjective motion of the surface vehicle platform and the camera platform, but also the motion caused by other detrimental external factors [3]. Fast visual-field movements may also cause motion blur [4]. In addition, more application scenarios, such as live broadcasting, further image processing, and so on, put forth higher requirements for the timeliness of the stabilization process. In general, the onboard video stabilization problem in heavy sea environments is a nonlinear problem with great complexity, timeliness, and strong coupling, and it has important theoretical significance and practical value [1].

Traditional video stabilization methods can be roughly divided into three kinds [5]. Mechanical image stabilization (MIS) keeps stable by compensating for the motion of the stabilization platform. Optical image stabilization (OIS) means keeping stable by compensating for optical paths [6]. Electronic image stabilization (EIS) keeps the visual field stable through image processing. Up to now, the technology of motion compensation with a gimbal is very mature and widely used by researchers [7,8,9,10]. In addition, many new gimbal control methods have been proposed to be adapted to different applications [11,12]. Cheng et al. designed a PID control method combined with a BP neural network, which has a short adjustment time, good tracking characteristics, a smooth transition process, and a small overshoot, and it can quickly meet control requirements [13]. Behzad et al. designed an inclusive dynamic model of a one-axis gimbal mechanism. The effectiveness of active disturbance rejection control (ADRC) in the presence of external disturbances and parameter uncertainties was illustrated [14]. Hao et al. proposed an optical stabilization system based on deformable mirrors (DMs) for retina-like sensors. This system achieved image stabilization by changing the reflective plate of the DM’s compensating tilt angle [15]. In particular, various corner detection algorithms were employed to accomplish corner detection and corner matching so as to obtain inter-frame motion estimations [1,3,16,17].

However, when tackling a more complex and ever-changing environment, a single stabilization method may not be adequate. A single mechanical stabilization method is readily affected by friction, wind force, sensor errors, etc. When upgrading technology, mechanical stabilization can mainly be improved by hardware optimization to ensure better stabilization effects, which requires a higher cost. Optical stabilization is mainly achieved through delicate and complicated optical anti-vibration devices. The accuracy of electronic stabilization often depends on the stability of the gray value or the accuracy of the feature point. As for heavy sea environments, feature points may only exist in possible landmarks or clouds in the sky, which may cause more misjudgment and loss [5]. The gray value of an image may also be unstable with changes in light conditions. Furthermore, an electronic stabilization method will lose a lot of information in the case of severe shaking of the visual field, making the method almost invalid. Hence, a hierarchical stabilization method is an effective way to solve these problems. Bayrak indicated that the combination of mechanical stabilization and electronic stabilization can achieve better results than those of a single stabilization method in the case of high amplitude jitter [18]. Windau and Itti presented a multilayer real-time video image stabilization system and suggested that different kinds of stabilization methods can interact and complement each other [19]. These results verify the rationality of the hierarchical stabilization control method.

In order to accomplish image stabilization with satisfactory timeliness and sensitivity in a heavy sea environment, a hierarchical stabilization method combining a mechanical stabilization method with an electronic stabilization method is proposed in this paper. A three-DOF gimbal with a robust controller is equipped with a camera, and the stabilization process is realized through real-time motion compensation for large-angle vibrations and electronic image stabilization. The main contributions can be summarized as follows:A hierarchical stabilization control method is proposed to cope with the complex environment of a heavy sea. A mechanical stabilization process is combined with an electronic stabilization process, which greatly improves the effectiveness and robustness.An electronic stabilization method based on sea–sky-line detection is proposed. In addition, an improved stabilization method of image clipping based on prior information is utilized. The experimental results reveal that the proposed method can effectively stabilize an image in a complex environment with noises of various frequencies superimposed. The algorithm can achieve more than 50 fps by clipping images according to prior information, which fully meets the real-time demand.

The remainder of this paper is structured as follows. The kinematic model of the three-DOF gimbal is described in Section 2, followed by the equation of the relationship between the kinematic parameters and the sea–sky line. The detailed experimental results are illustrated in Section 3. Conclusions and future work are in Section 4.

## 2. Hierarchical Stabilization Control Method

A visual flowchart of the proposed method is illustrated in Figure 1. The mechanical and electronic stabilization methods are combined in series, both of which are discussed below.

### 2.1. Kinematic Model

It is assumed that the surface vehicle is on the sea and the camera is carried on the surface vehicle. There is no object on the surface of the sea to refer to other than the sea–sky line. Considering the influence of atmospheric refraction on the surface, there is a conclusion of the relationship between θ and *h* [20]:(1)$$\theta =\sqrt{\frac{22h}{13r}}\approx 0.0295\sqrt{h},$$
where *r* is the radius of the earth, θ is the declivity angle, and *h* is the height from the sea level.

It is assumed that the effect of ocean waves on the size of *h* is very small relative to the radius of the Earth, in which case θ is approximately a fixed value. Thus, we can assume that *h* is zero. In other words, the camera is located near the sea level and fluctuates less with the waves. In this case, θ is always zero. The sea–sky line is in the center of the field of view. Therefore, the center is the original point of the sea–sky line. In addition, the other two translational motions of the camera along the surface of the sea obviously have no effect on the position of the sea–sky line in the field of view.

The relationship between the other three rotational degrees of freedom of the camera and the sea–sky line is described below. Let the reference frame be any fixed coordinate whose XOY plane coincides with the sea surface. Firstly, it is assumed that the orientation of the fixed coordinate of the camera, as shown in Figure 2, is obtained through a continuous relative rotation transformation of the base coordinate around its own *Z*-*X*-*Y* axis.
(2)p0=R10p1=R10R21p2=R10R21R32p3=R30p3,
where p0 is the value in the basis coordinate of any point at sea level, and Rmn is the rotation matrix of the *m*-th coordinate with respect to the *n*-th coordinate. Specifically, R10,R21,R32 are construed as
(3)R10=[cosα−sinα0sinαcosα0001],
(4)R21=[1000cosβ−sinβ0sinβcosβ],
(5)R32=[cosγ0sinγ010−sinγ0cosγ].

Thus, the rotation matrix of the camera can be described as
(6)R30=[cosα−sinα0sinαcosα0001][1000cosβ−sinβ0sinβcosβ][cosγ0sinγ010−sinγ0cosγ]=[cαcγ−sαsβsγ−cβsαcαsγ+sαsβcγsαcγ−cαsβsγcβcαsαsγ−cαsβcγ−cβsγsβcβcγ]
where α,β, and γ represent the angle of continuous relative rotation of the camera around the *Z*-*X*-*Y* axis with respect to the base coordinate, respectively, and s and c, respectively, represent sin and cos.

Note that p0 is the position of any point at sea level in the basis coordinate, which can be described as
(7)p0=[uv0]
where *u* and *v* can be any real number. Hence, it follows that
(8)p3=[u(cαcγ−sαsβsγ)+v(sαcγ+cαsβsγ)vcαcβ−usαcβu(cαsγ+sαsβcγ)+v(sαsγ−cαsβcγ)]
which indicates the representation of the sea–sky line in the fixed coordinate of the camera.

As can be seen from Figure 2, the angle between $X^{\prime }$ and the intersection line of the sea level and $X^{\prime }O^{\prime }Z^{\prime }$ can be expressed as
(9)$$\tan\theta =\frac{u(c\alpha s\gamma +s\alpha s\beta c\gamma )+v(s\alpha s\gamma -c\alpha s\beta c\gamma )}{u(c\alpha c\gamma -s\alpha s\beta s\gamma )+v(s\alpha c\gamma +c\alpha s\beta s\gamma )}=\tan\gamma$$
where θ is directly related to the intercept of the sea–sky line, as illustrated in Figure 3.

If the sea–sky line is represented in slope-intercept form, the view coordinate is established in the camera’s field of view, and the center of the field of view is taken as the origin, then the relationship between the intercept and the attitude angle can be described as
(10)$$b=\frac{N\sin\gamma }{2\sin\frac{\eta }{2}}$$
where *b* is the intercept of the sea–sky line, η is the field angle of the camera, and *N* is the pixel number of the height. Similarly, the slope of the sea–sky line in the field of view of the camera can be described as the intersection line of the sea surface and the $Y^{\prime }O^{\prime }Z^{\prime }$ plane.
(11)$$k=\frac{u(c\alpha s\gamma +s\alpha s\beta c\gamma )+v(s\alpha s\gamma -c\alpha s\beta c\gamma )}{vc\alpha c\beta -us\alpha c\beta }=-(\sin\gamma +\cos\gamma )\tan\beta$$
where *k* is the slope of the sea–sky line, as shown in Figure 4.

According to Equations (Equation 10) and (Equation 11), the attitude angles β and γ can be solved from *k* and *b*, i.e., the slope and intercept of the sea–sky line in the field of vision. Although the rotation of the yaw axis cannot be solved with sea–sky-line information, the sea waves usually cannot affect the yaw motion in real environments.

### 2.2. Mechanical Stabilization Method

Mechanical stabilization plays a vital role in image stabilization in heavy sea conditions. Firstly, mechanical stabilization can provide primary motion compensation for the entire amplitude spectrum, reducing the difficulty of the electronic stabilization algorithm [19]. In addition, it can also reduce the proportion of image compensation of the visual field in the electronic stabilization algorithm [21]. Secondly, the electronic stabilization method is limited by the exposure time and fps of the camera, which makes it difficult to deal with the high-speed motion of the camera itself or that of the objects. Mechanical stabilization can effectively reduce the possibility of motion blur in the visual field and improve the robustness of subsequent algorithms. Furthermore, the high-frequency noise brought by a wave or the motors on surface vehicles almost cannot be handled by simple electronic stabilization methods.

The simple BGC 32-bit three-axis gimbal control system developed by Basecam Electronics was used as the controller for the experimental data in this paper. The hardware platform used a three-axis stabilizer developed by Huizhou FOSICAM Technology. In this system, the traditional PID control method was used to realize the self-stability of the 3-DOF gimbal. It was equipped with a 6-DOF IMU placed in parallel next to the video camera. On this basis, a low-pass filter was utilized to stop the noise of high-frequency signals and reduce the possibility of high-frequency resonance. A superimposed trap filter could filter the noise with the specific frequency of the three-axis motors. By applying an adaptive-gain PID method, the PID parameters could be dynamically adjusted to make the moving process smoother. In order to improve the response speed in a heavy sea environment, the PID parameters of the three joint axes were adjusted first, and then the related parameters of adaptive gain were adjusted. The parameters of the adaptive-gain PID method included the RMS error threshold, attenuation rate, and recovery factor. RMS means that when the error exceeds this threshold, the adaptive PID algorithm starts to run. The larger the attenuation rate is, the more the proportional PID gain decreases. The recovery factor defines how fast the proportional PID gain recovers when the system becomes stable. The detailed steps of mechanical stabilization are shown in Figure 5.

### 2.3. Electronic Stabilization Method

Generally, the proposed electronic stabilization method includes preprocessing, sea–sky-line detection, motion compensation, and image compensation, as shown in Figure 6. The parts marked in yellow are the key contributions in this paper. Other parts have been extensively studied by researchers in past decades, and will not be covered here.

#### 2.3.1. Preprocessing

The proposed algorithm adopts the conventional electronic image stabilization preprocessing process, including Gaussian filtering, binarization, contrast adjustment, Canny edge detection, and other steps. Gaussian filtering can make the result of edge detection less affected by wave fluctuation. In the contrast adjustment process, linear contrast adjustment is used to achieve better edge detection results and make it more robust. The Canny edge detection algorithm can calculate the gradient image, get the single-pixel edge line after non-maximum suppression, and then get the binary image with the selected sea–sky line through double-threshold processing and edge stitching [22].

#### 2.3.2. Sea–Sky-Line Detection

For a binary image of several candidate sea–sky lines with continuous pixels, the Hough transform is used to get the real sea–sky line. The classical Hough transform can map points on the same line in the original image to a point in Hough space, forming a peak value. Then, one can set a threshold value to extract the line that is most likely to be the sea–sky line in the image [23]. If multiple sea–sky lines are extracted with the Hough transform, the outliers are removed from these lines according to the prior information. Then, the average value of the slope and intercept of the remaining sea–sky line will be taken as the final result.

#### 2.3.3. Interframe Motion Estimation

The process of motion estimation involves obtaining the kinematic parameters of the camera according to the relationship of the slope and intercept of two consecutive sea–sky lines. According to Equations (Equation 10) and (Equation 11), it follows that
(12)$${\dot{\gamma }}_{T}=f(\arcsin(\frac{2b_{T}}{N}\sin\frac{\eta }{2})-\arcsin(\frac{2b_{T-1}}{N}\sin\frac{\eta }{2}))$$
(13)$${\dot{\beta }}_{T}=f(\arctan\frac{-k_{T}}{\sin\gamma +\cos\gamma }-\arctan\frac{-k_{T-1}}{\sin\gamma +\cos\gamma })$$
where *f* is the reciprocal of the time between two frames, which indicates the fps, and *T* is a counting variable.

#### 2.3.4. Image Clipping Based on Prior Information

The image preprocessing step also includes image clipping based on prior information with the aim of reducing the amount of data and improving timeliness according to prior information. The clipped image should contain the region of the sea–sky line in the previous frame and the possible interframe motion, plus an additional error redundancy, as shown in Figure 7. Therefore, the formula of the remaining pixel height is as follows:(14)$$h_{T}=|k_{T}|W+\sqrt[{}]{2}\;|\tan\beta_{T}-\tan\beta_{T-1}|\;W+\delta$$
where *W* is the width of the frame and δ is the additional error redundancy, which is set to 20 in this paper.

## 3. Results and Analysis

### 3.1. Results of the Mechanical Video Stabilization Experiment

For mechanical stabilization, the response of the three-DOF gimbal to large-angle sine signals is demonstrated by comparing the angle data, which were measured by the IMUs attached to the pedestal and camera. In this experiment, a Stewart platform was used as the source of external vibrations to simulate the influence of wave fluctuations on camera motion [24]. The Stewart platform was set to move with only three degrees of rotational freedom as a sine function. The rotation amplitude was 15${}^{\circ }$ and the frequency was 0.5 Hz, which simulated the violent vibrations in a heavy sea environment. Two IMUs were attached to the Stewart platform to measure the angular displacement. Figure 8 shows the experimental platform of mechanical stabilization at different times. The camera could basically maintain a stable attitude when the Stewart platform moved.

Figure 9 plots the pitch angle measured on the Stewart platform and the stabilized platform at 0.5 Hz. The mechanically stabilized motion of the three degrees of freedom was approximately a sine motion. Figure 10 demonstrates the amplitude of the angle displacement of the three degrees of freedom. At 0.5 Hz, the large-angle mechanical vibration is reduced by 74.2% on average. Therefore, it is noticeable that mechanical stabilization can effectively reduce the amplitude and velocity of large-angle motion and reduce the motion compensation and image compensation of electronic stabilization. In addition, since mechanical stabilization has been proven to be effective in improving image quality [25], it can also improve electronic stabilization. Better image quality is beneficial in improving the success rate of the following sea–sky-line detection.

### 3.2. Results of the Electronic Video Stabilization Experiment

Figure 11 demonstrates the result of each step in the sea–sky-line detection algorithm. The proposed electronic video stabilization algorithm had a good effect on the visual environment of the sea–sky line. Remarkably, according to the time cost analysis of each step of the algorithm, the electronic video stabilization method based on image clipping could effectively reduce the processing time. Note that the tests were conducted on a 2.5 GHz i7 processor with 16 GB of RAM. For a 1080 p input video, the average processing frame rate could reach 50 fps, which was able to meet the real-time processing requirement of a camera with a frame rate of 30 fps.

In the meantime, the center of the visual field was considered as the origin of motion compensation of the sea–sky line. Some representative frames (30th, 60th, 90th, and 120th) of the video were selected for display, as shown in Figure 12. In particular, the mean squared error (MSE) was imported to describe the change in intensity between two images, as shown in Equation (Equation 15). The results of the MSE analysis are plotted in Figure 13. As can be readily identified, the mean value of the MSE declined from 108.91 to 60.94, corresponding to a decreasing amplitude of approximately 42.1%.
(15)$$MSE=\frac{1}{M\times N}\sum_{i=0}^{M-1}\sum_{j=0}^{N-1}{\left[I\left(i,j\right)-K\left(i,j\right)\right]}^{2}$$

Furthermore, a Windows system API in a C++ environment was called to measure the execution time of each step of the electronic stabilization algorithm with an accuracy of microseconds. The execution time of each step of the image clipping based on prior information in comparison with the original algorithm is shown in Figure 14. As can be observed, the image-clipping method based on prior information was able to effectively reduce the time taken for electronic stabilization by 14.9%. Except for the step of Gaussian filtering, the time consumption of the other steps was able to be effectively reduced.

In order to further verify the robustness of the algorithm, we tried to find more surface videos in different environments and conditions to test the algorithm. Video 1 has strong sunlight reflections on the surface. As an external interference factor, strong sunlight will cause the interruption of large gradients at the sea–sky line, reducing the success rate of sea–sky-line detection. Video 2 has a mountainous background, which may affect the sea–sky-line detection. Video 3 was shot in harsh conditions. Heavy waves made it impossible for the sea–sky line to stay in line. Fog made the sky hazy and blurred the sea–sky line. Due to the lack of mechanical stabilization, the field of vision shook violently. The original videos are posted on Google Drive (https://drive.google.com/file/d/11iHkR5ggZA1IuXHvCTURpz0XIGFR2BYt/view?usp=sharing (accessed on 28 January 2022)) for readers who need them.

Figure 15 shows the effects of electronic stabilization in different environments. Clouds and sunlight had little effect on the proposed electronic stabilization algorithm, as shown in Figure 15a,b. Our algorithm was also able to distinguish between the sea–sky line and the mountain line in the background, as shown in Figure 15c. Storms at sea can really affect the quality of videos at sea. Even so, the algorithm still had 92.4% success rate of sea–sky-line detection. Table 1 lists the success rates of sea–sky-line detection for the marine videos with different interference factors.

### 3.3. Discussion

In brief, the proposed method can meet the requirements of image stabilization in a heavy sea environment. The combination of mechanical stabilization with large-angle motion compensation and the maritime electronic stabilization method with high time efficiency and robustness can achieve a better effect than a single method can. At present, military and civilian activities on the sea are becoming more and more frequent. The proposed method can be widely applied to surface vehicles to enhance the quality of video at sea, and then further improve the accuracy of the acquired information, which has broad application prospects. Therefore, it is of great commercial value for managers of companies engaged in offshore operations. Stable offshore video is good for post-processing and applications such as target identification and tracking. Although the proposed method of the hierarchical stabilization method is more expensive, as it requires the hardware of a three-DOF gimbal and processor, it is still superior to pure electronic stabilization in heavy sea environments.

However, due to the limitations of the experimental site and time, the proposed method was not further verified in more application scenarios. The main disadvantage of the electronic stabilization method proposed in this paper is the lack of robustness in scene switching. According to the current research, this method is suitable for the maritime video environment, but probably not for other environments where the sea–sky line does not exist. In addition, detecting a sea–sky line incorrectly over a period of time or not detecting it may result in poor stabilization performance. These are all problems that might be worth investigating further.

## 4. Conclusions and Future Work

In this work, a novel hierarchical image stabilization method was proposed and implemented for a three-axis gimbal mounted on surface vehicles; it combines mechanical stabilization with electronic stabilization. Specifically, the mechanical stabilization method can effectively reduce the vibration amplitude and filter high-frequency noise, thus offering better conditions for electronic stabilization algorithms. The electronic stabilization method based on sea–sky-line detection can detect the sea–sky line in real time, calculate the attitude angle, obtain the interframe motion model, and then compensate for the motion. The obtained results demonstrate that the proposed stabilization control method can satisfy both the requirements of robustness and real-time operation in heavy maritime environments.

Ongoing and future work will involve two aspects. On the one hand, according to the characteristics of heavy sea environments, we will design another small three-DOF gimbal to obtain better controllability and anti-interference ability compared with those of common hand-held gimbals. On the other hand, for the electronic stabilization algorithm, the current method based on sea–sky-line detection often fails to utilize other stable features than the sea–sky line. An urgent task is to optimize the effects of electronic stabilization by effectively utilizing other stable features apart from the skyline in a sea–sky scenario.

## Figures and Tables

**Figure 1 sensors-22-02587-f001:**
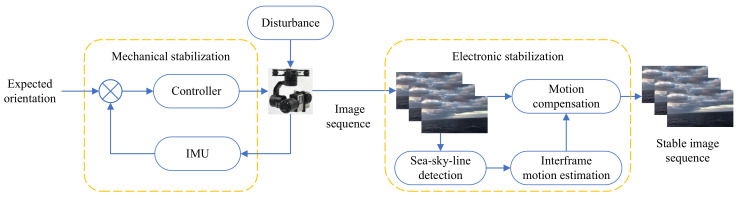
Flowchart of hierarchical video stabilization control method.

**Figure 2 sensors-22-02587-f002:**
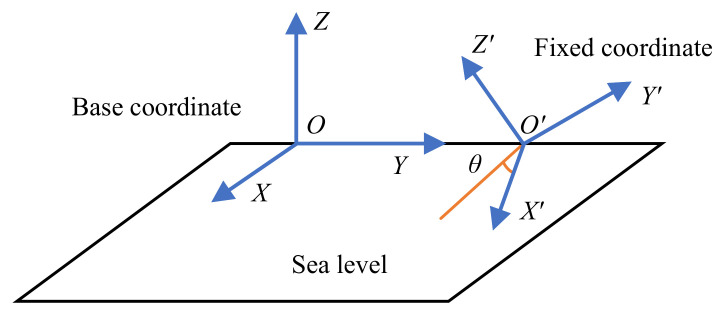
Kinematic model of the camera. The fixed coordinate is obtained through a continuous relative rotation transformation of the base coordinate.

**Figure 3 sensors-22-02587-f003:**
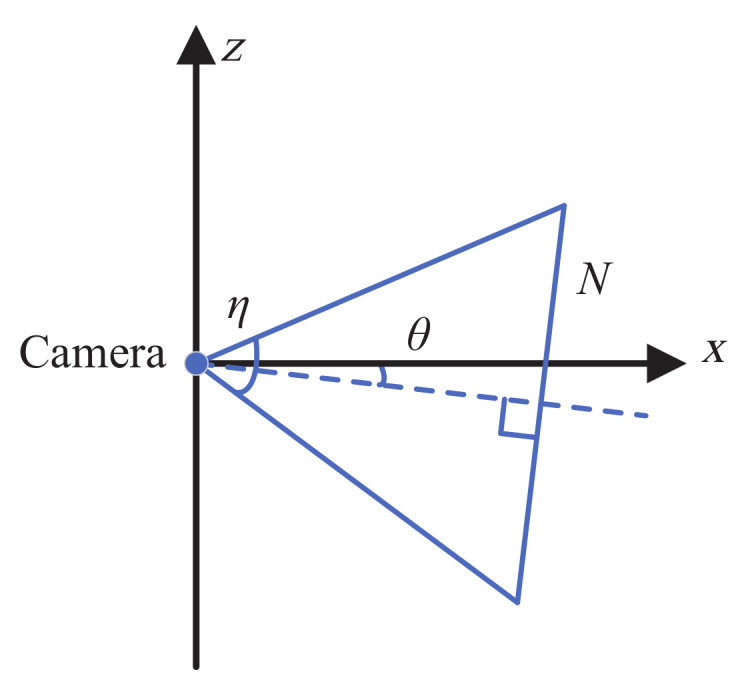
Relationship between θ and the intercept of the sea–sky line.

**Figure 4 sensors-22-02587-f004:**
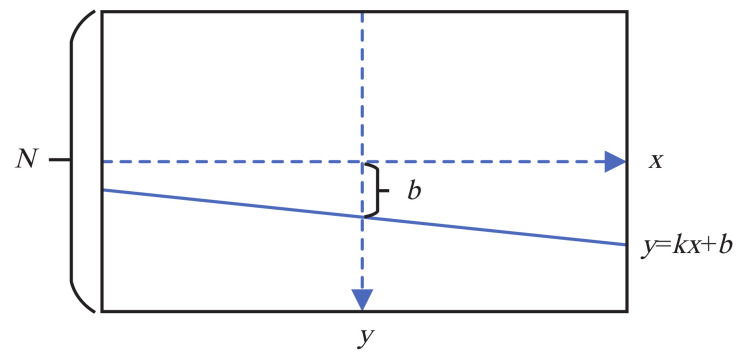
Sea–sky line in slope-intercept form.

**Figure 5 sensors-22-02587-f005:**
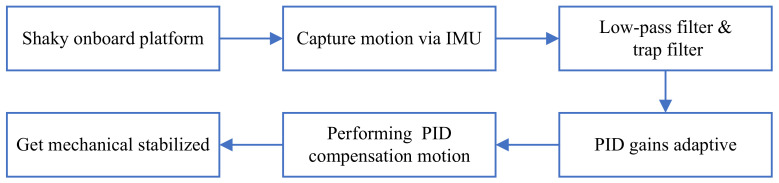
Detailed steps of mechanical stabilization.

**Figure 6 sensors-22-02587-f006:**
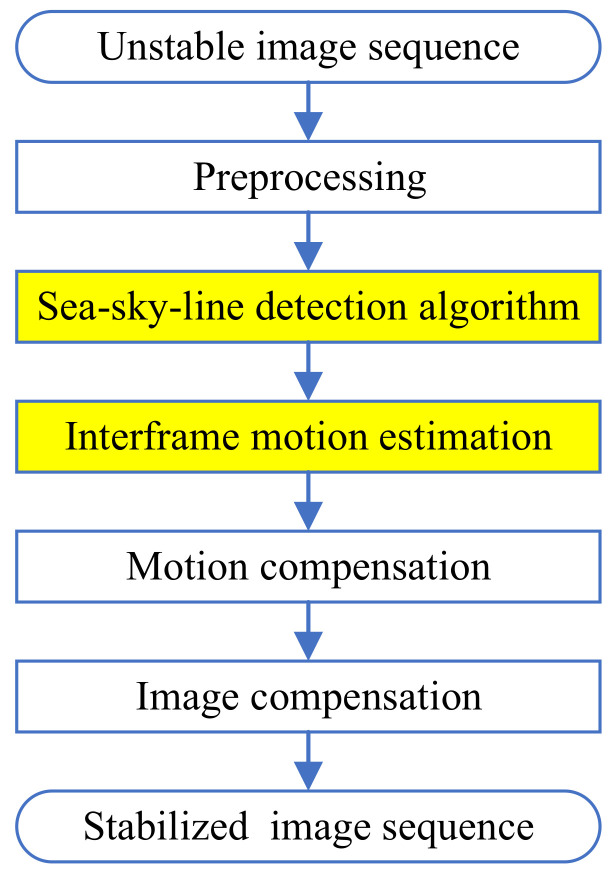
Flowchart of the electronic stabilization.

**Figure 7 sensors-22-02587-f007:**
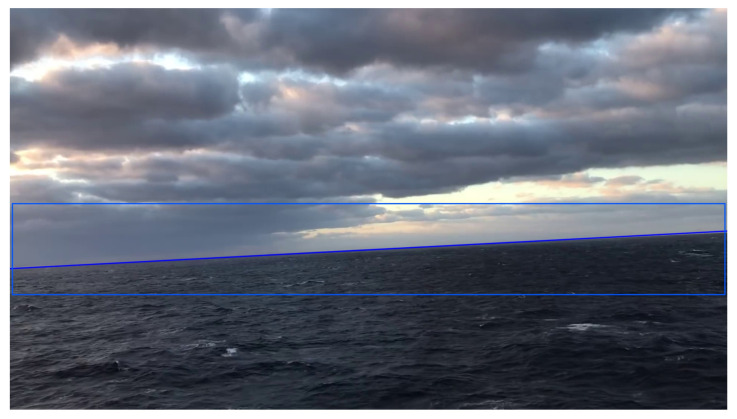
Clipped image based on prior information.

**Figure 8 sensors-22-02587-f008:**
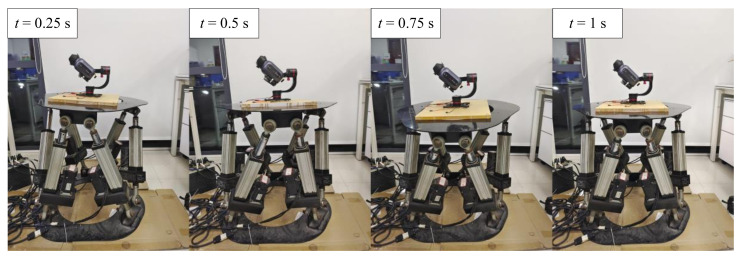
Experimental platform for mechanical stabilization at different timestamps.

**Figure 9 sensors-22-02587-f009:**
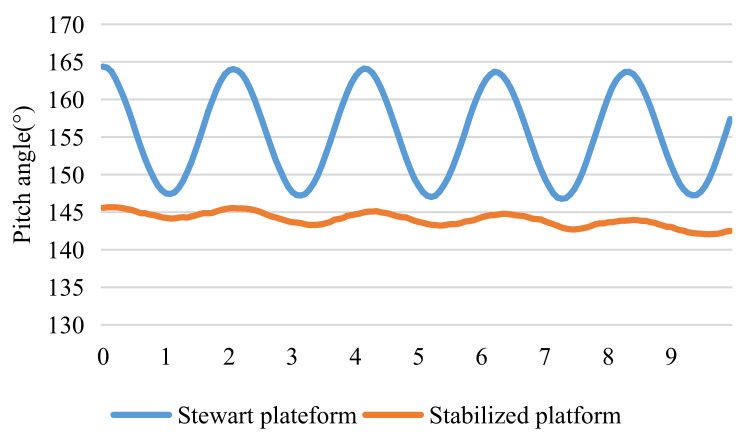
Motion compensation effect of mechanical stabilization.

**Figure 10 sensors-22-02587-f010:**
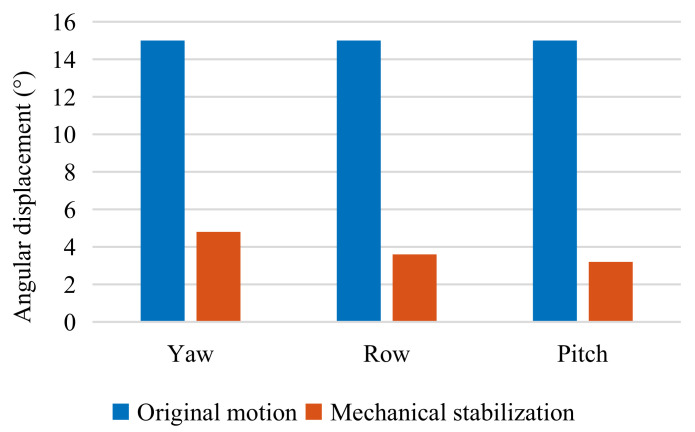
Motion amplitudes before and after motion compensation through mechanical stabilization in three rotating directions.

**Figure 11 sensors-22-02587-f011:**
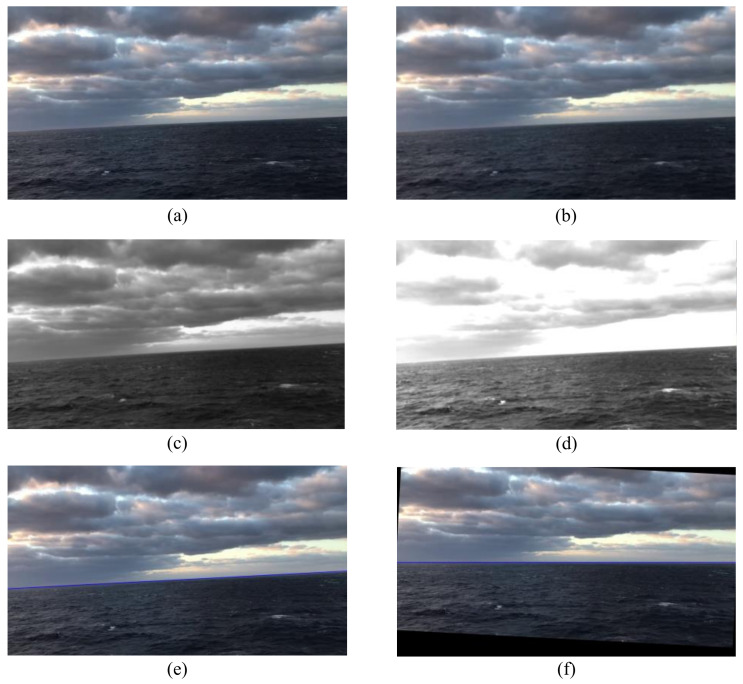
Process of electronic video stabilization. (**a**) The original picture; (**b**) Gaussian filtering; (**c**) gray-level image; (**d**) contrast adjustment; (**e**) Canny edge detection; (**f**) motion compensation.

**Figure 12 sensors-22-02587-f012:**
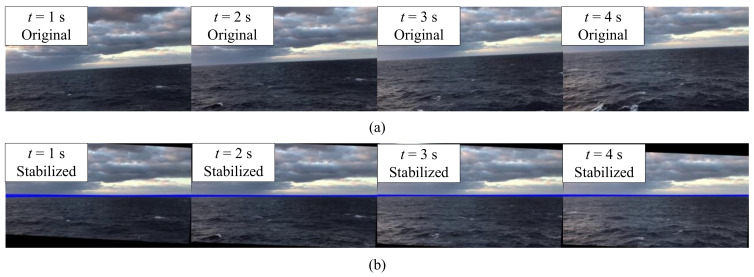
Comparison of the original and stabilized images from the electronic stabilization algorithm. (**a**) The original image; (**b**) the stabilized image.

**Figure 13 sensors-22-02587-f013:**
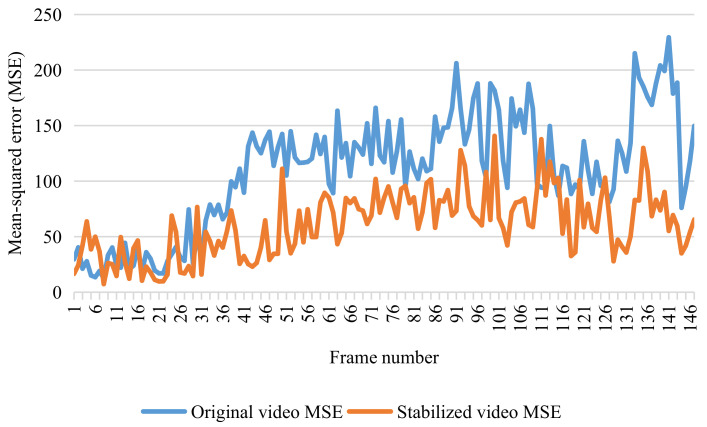
Mean squared error of the original video and the stabilized video.

**Figure 14 sensors-22-02587-f014:**
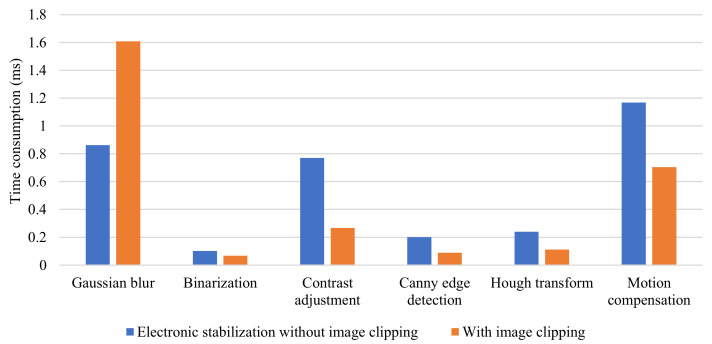
Processing time of each step of electronic stabilization.

**Figure 15 sensors-22-02587-f015:**
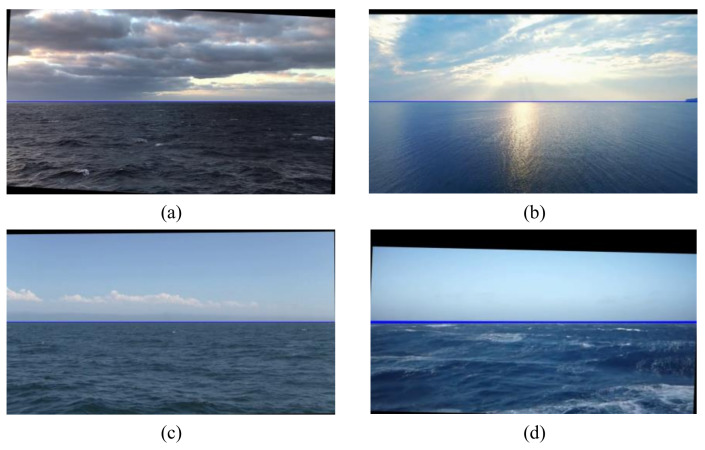
The effects of electronic stabilization in different environments: (**a**) in cloudy weather conditions; (**b**) the sunlight reflecting heavily on the sea surface; (**c**) a mountainous background; (**d**) in a severe storm.

**Table 1 sensors-22-02587-t001:** The success rate of the electronic stabilization algorithm in different environments.

Video Number	Success Rate	Experiment Frame Number
Original video	98.6%	2000
Video 1	99.1%	400
Video 2	97.7%	1000
Video 3	92.4%	2000

## Data Availability

Data can be made available upon request from the corresponding author.

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
