# Peer review of "A Hierarchical Stabilization Control Method for a Three-Axis Gimbal Based on Sea–Sky-Line Detection"

_sensors, 2022, doi:10.3390/s22072587_

Round 1

Reviewer 1 Report

This paper mainly describes a novel method of video stabilization in heavy sea environments. Compared with the common single electronic or mechanical stabilization method, it puts forward a hierarchical stabilization method to deal with more complex and changeable environment, where the video is shoot onboard with severe disturbances. As can be summarized, the main contribution of this paper is proposing a hierarchical stabilization method based on 3-DOF gimbal and sea-sky-line detection. The reviewer suggests accepting this paper after minor revision.
More minor points are listed below.
1. In terms of the structure, the content of the second and fourth paragraphs is partly repeated, which are both about the problems of the traditional stabilization method. It is advisable to combine them together or have another way to express.
2. There is a lack of overview on other hierarchical stabilization methods.
3. When discussing about the method, the absence of pictures of the experiment platform will make it difficult for readers to understand the meaning expressed by author. It is recommended to add an actual or model image of the gimbal.
4. There are some typos in this paper, please check and correct them carefully.

Author Response

For the point-by-point responses to the comments, please see the attached response file. Many thanks!

Reviewer 2 Report

I have the following comments and questions for the authors to address:

  1. Some significant finding should be mentioned in the abstract
  2. More critical review should be added to the paper with reference to latest published papers
  3. A more detail discussion on the result presented is needed
  4. Detail comparison with previous research work should be presented in the form of a table with specific parameters
  5. Highlight the limitation of the proposed method is needed
  6. The paper requires editing
  7. Authors should provide more information about their specific assumptions. This is very important as anyone in the world can reproduce the provided results easily at any time.
  8. Authors should provide validation about the presented special cases as they said. Further, how can you compare and validate your results with other cases published in literature. More tips please such as the reason for producing NEW analysis for the developed results, old and new work in with it, what can we learn and hope from it in the future.
  9. The sensitivity analyses on the key parameters of this process is not complete.
  10.  Where are the managerial insights? What are the main findings for industries? How this paper can help the managers? What are the practical solutions for the proposed problem in real domain?
  11. The authors have to indicate the error propagation about their algorithms to estimate the solutions.
  12. The authors have to show the cross validation into the training and testing

Author Response

(The authors gave the same response as above.)

Reviewer 3 Report

  1. Is the Figure 9 (c) is Binarization (misspelled)? It looks like gray level.
  2. The authors stated the mechanical stabilization can reduce the possibility of motion blur. The experiment result just shown the Angular displacement in Figure 8, and didn’t compare the image distortion under the conditions of with and without mechanical stabilization.

  3. The authors just implemented the proposed method on Steward platform. However, general real marine environment has more severe and unknown disturbance and affect the stability of vessels more seriously. Moreover, vessel control under the real environment is more challenging.
    I suggest the authors should implement their method on a real aquatic environment, and verify that using a real vessel.

Author Response

(The authors gave the same response as above.)

Round 2

Reviewer 2 Report

Accept the paper

Author Response

(The authors gave the same response as above.)

Reviewer 3 Report

Maybe the revised time is so short, so that the authors cannot respond well to the reviewer's comments.

Even if the motion platform can simulate the motion of ships, the marine environment is often harsh, adverse weather conditions including rain, fog and haze will affect the visibility and reduce the quality of the image.

I think the authors should have more experiments to prove that the proposed method can be applied in the actual marine environment.

1. The caption of Figure 11 (c) is still misspelled (iamge??).

2. The motion blur for imaging on the sea may not be a serious problem, because the relative speed of the camera to the environment is slow. Then, the effect of the mechanical stabilization is limited.

I don't think the Figure 9 and Figure 10 can present the effectiveness of the mechanical video stabilization. They just showed the mechanical angles not the comparison of the image quality.

If the image quality doesn't change, the mechanical video stabilization is not necessary.

3. I think the virtual environment cannot simulate the ill-condition on the sea which degrades the image quality seriously. The simulated environment and condition is far from the real marine environment.

Author Response

(The authors gave the same response as above.)

Round 3

Reviewer 3 Report

1.

I still think that it is not enough to have no ocean experiments, because the mechanical stabilization is not been verified by the actual sea conditions. But I can also understand the difficulty and high cost of actual ocean experiments. Hence, it's a pity for this paper.

2.

There are typographical and grammatical errors throughout the paper, especially the revised content.  It should be corrected before the paper is accepted.

Ln 265 "The lines of the mountains in view are often not straight enough to be mistaken for .."  What does it mean?

Some Chinglish writing is not easily understood. The English must be proofread by native speakers.

Author Response

(The authors gave the same response as above.)
